# NCX-4040, a Unique Nitric Oxide Donor, Induces Reversal of Drug-Resistance in Both ABCB1- and ABCG2-Expressing Multidrug Human Cancer Cells

**DOI:** 10.3390/cancers13071680

**Published:** 2021-04-02

**Authors:** Birandra K. Sinha, Lalith Perera, Ronald E. Cannon

**Affiliations:** 1Laboratory of Toxicology and Toxicokinetic, National Cancer Institute at National Institute of Environmental Health Sciences, Research Triangle Park, NC 27709, USA; cannon1@niehs.nih.gov; 2Laboratory of Genome Integrity and Structural Biology, National Institute of Environmental Health Sciences, Research Triangle Park, NC 27709, USA; pereral2@niehs.nih.gov

**Keywords:** NCX4040, nitric oxide-donors, nitric oxide, P-gp protein, breast cancer resistance protein, adriamycin, topotecan

## Abstract

**Simple Summary:**

Development of resistance to chemotherapeutics during the treatment of human cancers is a serious problem in the clinic, resulting in a poor treatment outcome and survival. It is believed that overexpression of ABC efflux proteins (e.g., P-gp/ABCB1, BCRP/ABCG2 and MRP/ABCC1) on the tumor cell membrane is one of the main mechanisms for this clinical resistance. Our recent studies indicate that nitric oxide (NO), inhibits ATPase functions of ABC transporters, resulting in reversal of resistance to various anticancer drugs. In this study we have found that nitric oxide and/or active metabolite (s) generated from NCX4040, a nitric oxide donor, inhibited ABC transporter activities by inhibiting their ATPase functions, causing reversal of both adriamycin and topotecan resistance in human MDR tumor cells. We also found that nitric oxide and/or metabolites of NCX4040 significantly enhanced drug accumulations in MDR tumor cells. These studies strongly suggest that tumor specific nitric oxide donors that deliver high amounts of nitric oxide and reactive species to clinical resistant tumors may be extremely useful in treating human tumors overexpressing ABC transporters, including cancer stem cells. In addition, such NO-donors can also be utilized for treating other diseases where drug resistance results from the presence of ATP-dependent transporter proteins.

**Abstract:**

The emergence of multidrug resistance (MDR) in the clinic is a significant problem for a successful treatment of human cancers. Overexpression of various ABC transporters (P-gp, BCRP and MRP’s), which remove anticancer drugs in an ATP-dependent manner, is linked to the emergence of MDR. Attempts to modulate MDR have not been very successful in the clinic. Furthermore, no single agent has been found to significantly inhibit their functions to overcome clinical drug resistance. We have previously shown that nitric oxide (^●^NO) inhibits ATPase functions of ABC transporters, causing reversal of resistance to clinically active anticancer drugs. In this study, we have used cytotoxicity and molecular docking studies to show that NCX4040, a nitric oxide donor related to aspirin, inhibited the functions of ATPase which resulted in significant reversal of resistance to both adriamycin and topotecan in P-gp- and BCRP-expressing human cancer cell lines, respectively. We also used several other cytotoxic nitric oxide donors, e.g., molsidomine and S-nitroso glutathione; however, both P-gp- and BCRP-expressing cells were found to be highly resistant to these NO-donors. Molecular docking studies showed that NCX4040 binds to the nucleotide binding domains of the ATPase and interferes with further binding of ATP, resulting in decreased activities of these transporters. Our results are extremely promising and suggest that nitric oxide and other reactive species delivered to drug resistant tumor cells by well-designed nitric oxide donors could be useful in sensitizing anticancer drugs in multidrug resistant tumors expressing various ABC transporters.

## 1. Introduction

The phenomenon of multidrug resistance (MDR) remains a major challenge for the treatment of human cancers in the clinic. Development of resistance to chemotherapeutics during the treatment of is a serious problem as they lead to poor outcome and death. While various mechanisms are responsible for resistance of cancer cells to cytotoxic drugs, overexpression of ABC efflux proteins (e.g., P-gp/ABCB1, BCRP/ABCG2 and MRP/ABCC1) on the tumor cell membrane play a significant role in the emergence of resistance to antineoplastic drugs [1,2,3]. These ATP-dependent transporters remove various anticancer drugs from tumor cells, effectively reducing the drug concentrations from cellular targets, resulting in tumor cell survival [3,4]. While a significant amount of research now has identified various compounds that inhibit functions of these ABC transporters in vitro, their utility in the clinic have not been very successful due to significant toxicity associated with these compounds [5,6]. It should be noted that ABC transporters play a dual role as they are also highly expressed in normal tissues, causing in alterations of the drug absorption, distribution, metabolism and removal [7,8], while in tumor cells overexpression of these transporters results in active efflux of drugs [9].

Studies from our laboratory [10,11] as well as others [12] have identified nitric oxide donors as anticancer drug sensitizers which are capable of reversing resistance mediated by ABC transporters in human MDR tumor cells. Nitric oxide donors release nitric oxide (^●^NO) in cells which is a short-lived free radical molecule and acts as a cellular messenger [13]. ^●^NO is involved in many important physiological functions and has been shown to cause killing of various tumor cells in both in vitro and in vivo [14]. Many of the actions of ^●^NO results from the formation of various reactive metabolites (e.g., NO^+^, N_2_O_3_, and –OONO^●^) in cells. These intermediates react with sulfhydryl groups of proteins, generating S-nitrosothiol-protein which results in the modulation of both the activity and stability of proteins [15]. High activity of iNOS, the inducible form of nitric oxide synthase, has been shown to be present in human tumors, generating significant amounts of NO in these tumors [16,17].

Our previous studies have shown that ^●^NO, delivered by NO-donors, inhibits ATPase of activity of both topoisomerase II [18] and ATP-dependent ABC transporters in human MDR tumor cells [10,11]. This inhibition of the ATPase activity of transporter proteins results in significant reversals of adriamycin, topotecan and mitoxantrone resistance in human MDR cells [10,11]. NO-donors, in general, are not tumor-specific and therefore, do not specifically deliver significant amounts of ^●^NO in tumors. Therefore, in our earlier study we utilized JS-K, a tumor specific NO-donor, activated by GSH/GSH transferase system in tumors [19], and found that JS-K was extremely effective in reversing P-gp-mediated resistance of ADR in a P-gp-overexpressing human tumor cell line. However, JS-K was ineffective in reversing drug resistance in a BCRP-overexpressing tumor cell line [11].

As MDR tumors in the clinic also overexpress more than one ABC transporters (e.g., P-gp, BCRP and MRP’s), and that modulators of ABC transporters are toxic, one cannot utilize different inhibitors of each transporters in combinations with anticancer agents without significantly increasing toxicity. Thus, it is of paramount importance to design one single compound which can effectively inhibit all of these ABC transporters at a subtoxic dose. As ^●^NO inhibits ATPase activities of ABC transporters, and ATPase functions are essential for hydrolysis of ATP to provide the energy necessary for transporters to function. We believe that ATPase domains of ABC transporters are ideal inhibitable targets, therefore we have continued our search and design of unique NO-donor (s) which may accomplish this successfully in the clinic.

We report here that NCX4040 (Figure 1) is effective in inhibiting the ATPase activities of both P-gp- and BCRP-expressing human tumor cells which resulted in significant reversals of resistance to both adriamycin (P-gp) and topotecan (BCRP) in MDR tumor cells. NCX4040, a non-steroidal anti-inflammatory drug NO-donor, is a nitro derivative of aspirin and has been shown to release ^●^NO following hydrolysis in cells. It is significantly cytotoxic to a number of tumor cells both in vitro and in vivo [20,21,22]. It has been shown to sensitize cis-platin against resistant tumors [23]. In addition to NCX4040, we have also used several other cytotoxic NO-donors (Figure 1), including molsidomine (MS), its metabolite linsidomine (SIN-1), and S-nitroso glutathione (GSNO). We have also carried out cytotoxicity and molecular docking studies with these NO-donors in order to further identify a lead NO-donor compound (s) that may prove to be significantly better in reversing multidrug resistance mediated by ABC transporters in the clinic.

## 2. Materials and Methods

Adriamycin (ADR) was a gift of the Drug Synthesis and Chemistry Branch, Developmental Therapeutic Program of NCI, NIH and Topotecan HCl (TPT) was purchased from Cayman Chemicals (Ann Arbor, MI, USA). Stock solutions of ADR and TPT were prepared in double distilled water and stored at −80 °C. In some cases, TPT was also dissolved in DMSO, and stored at −80 °C. The Hoechst 33342 dye (Thermo Fisher Scientific, Waltham, MA, USA) was dissolved in water. S-nitroso glutathione (GSNO), molsidomine (MS), SIN-1. HCl (SIN-1) and 2-(acetyl)benzoic acid 4-(nitroxymethyl) phenylester (NCX4040) were obtained from Sigma Chemicals (St. Lois, MO, USA). Stock solution of NCX4040 was prepared in DMSO and stored at −80 °C. Fresh drug solutions prepared from stock solutions were used in all experiments. Molsidomine was dissolved in DMSO and the fresh solutions of all other NO-donors were prepared and used immediately. Primary antibody for the detection of P-gp, BCRP and β-actin were obtained from Abcam (Cambridge, MA, USA).

### 2.1. Cell Culture

Human ovarian OVCAR-8 and NCI/ADR-RES cells, selected for resistance to adriamycin [24,25,26] were obtained from the NCI-Frederick Cancer Center (Frederick, MD, USA), and human MCF-7 breast tumor cells were obtained from ATCC (Manassas, VA, USA). MCF-7/MXR breast tumor cells, selected for resistance by exposure to mitoxantrone (MX) as described before [27], was a gift of Dr. Erasmus Schneider. Cells were cultured in Phenol Red-free RPMI media supplemented with 10% fetal bovine serum and antibiotics (complete media). Cells were routinely used for 20–25 passages, after which the cells were discarded, and a new cell culture was started from fresh, frozen stock.

### 2.2. Cytotoxicity of Nitric Oxide Donors in Tumor Cells and Reversal of Resistance

For the cytotoxicity studies 100,000–150,000 cells/well were plated in 2 mL of complete medium onto a 6-well plate (in duplicate) and allowed to attach for 18 h as described previously [18]. Cells (OVACAR-8, NCI/ADR-RES, MCF-7 and MCF-7/MXR) were treated with various concentrations of NO-donors (GSNO, MS, SIN-1 or NCX4040) for 72 h in complete medium (containing 200 µL of PBS, pH 7.0, 2 mL). Cells were trypsinized, and the numbers of surviving cells were determined by counting cells in a cell counter (Beckman, Brea, CA, USA). DMSO (0.01–0.1%) was included as the vehicle control when used.

For the reversal of resistance, cells were first treated with NCX4040 for 2 h in complete medium (containing 200 µL of PBS, pH 7.0, 2 mL) followed by the addition of various concentrations of drugs (ADR or TPT) and cells were further incubated for 72 h in complete medium. We used NCX4040 as our NO donor for drug resistance reversal studies as other NO-donors were found to be either not effective (required very high concentrations) or were substrates for P-gp and/or BCRP.

### 2.3. Effects of NCX4040 on P-gp and BCRP Proteins Levels by Western Blot Assay

The analyses for expression of P-gp and BCRP proteins and effects of ^●^NO/NCX4040 on levels of proteins were examined in cells by the Western Blot method. Cells (NCI/ADR-RES and MCF-7/MX) were treated with NCX4040 (5 µM) for different times (4, 24 and 48 h). Cells were collected following trypsinization, washed (2×) with ice-cold PBS and frozen at −80 until analysis. Cell pallets were thawed and lysed with lysing buffer containing inhibitors of proteolytic enzymes according to the manufacturing protocols. 10 μg of membrane proteins was electrophoresed for 50 min at 200 volts under reducing conditions on 3–8% Tris-acetate gels (Novex, Life Technologies, Carlsbad, CA, USA). Proteins were transferred onto BDVF membranes following electrophoresis. The membrane was incubated at room temperature for 30 min in 50 mL blocking buffer (Intercept^®^ Blocking Buffer, Licor, Lincoln, NE, USA), washed with 1× PBS and hybridized overnight at 4 °C in 1× PBS containing 0.1% Tween with 1:200 vol/vol P-gp (Abcam; catalog no. 170904) and BCRP (Abcam; catalog no. 207732) and 1:5000 vol/vol β-actin (Abcam; catalog no. 8224) primary antibodies. Excess antibodies were removed by washing the membrane three times with 1× PBS containing 0.1% Tween and incubated at room temperature for 1 h in 1× PBS with 1:10,000 vol/vol secondary antibodies [Odyssey Goat anti-Mouse IR Dye 800 CW or Goat anti-Rat IR dye 680 CW (Licor)]. The unbound secondary antibodies were then removed by washing three times with 1× PBS-0.1% Tween. An Odssey infrared imaging system (Li-Cor Biosciences, Lincoln, NE, USA) was used to acquire images.

### 2.4. Effects of NCX4040 on Accumulation of Adriamycin in P-gp-Expressing Cells

Accumulation of ADR in OVCAR-8 and NCI/ADR-RES cells were carried out as described previously [11,28]. Briefly, about 100,000 cells were seeded in six-well cover slips for 18 h in complete medium. After 18 h, the medium was removed, and 2 mL of fresh media was added. Cells were preincubated with different concentrations of NCX4040 or verapamil (10 µM) for 2 h. Then, ADR (5 µM) was added, and the cells were incubated for 2 h, washed with ice-cold PBS (2×), kept in ice-cold PBS at ice and examined using confocal microscopy. Average luminal fluorescence intensity of each cell was quantified using ImageJ software. 20–25 images were taken for each treatment group and controls, and 20≤ cells per group were analyzed. The means of fluorescence intensity were analyzed using GraphPad Prism (GraphPad Software, Inc, La Jolla, CA, USA). The data were subjected to one-way ANOVA and multiple comparisons.

### 2.5. Effects of NCX4040 on Accumulation of Hoechst 33342 in BCRP-Expressing Cells

Accumulations of Hoechst 33342 in MCF-7 and MCF-7/MX cells were carried out as described previously [11,29]. Briefly, about 100,000 cells were seeded in six-well cover slips in complete medium. After 18 h, the medium was removed, and 2 mL of fresh media was added. Cells were preincubated with various concentrations of NCX4040 for 2 h, followed by the addition of Hoechst 33342 (5 × 10^−7^ M) and incubated for additional 2 h. Cells were then washed with ice-cold PBS (×2), kept on ice in 1 mL of PBS, and examined using confocal microscopy.

### 2.6. Molecular Modeling Studies

Molecular modeling studies were carried out as described previously [11]. Briefly, structures of both P-gp (pdb ID: 6C0V) and BCRP (pdb ID: 6HBU) proteins were considered in receptor-ATP-bound conformations. The ligand binding site of proteins were prepared for docking using the Make_Receptor module (Release 3.2.0.2) with the help of bound ATP as the ligand. The HYBRID module (Release 3.2.0.2) of the OpenEye software package (OpenEye Scientific Software, Inc., Santa Fe, NM, USA; www.eyesopen.com) was used for docking studies. Omega2 module was utilized to construct all possible conformations of NO-donors (GSNO, SIN-1 and NCX4040). The scorings were done using the default ChemGauss4 scoring function of the HYBRID module and the scoring function is based on the shape of the ligand, hydrogen bonding between ligand and receptor, hydrogen bonding interactions with implicit solvent, and metal-chelator interactions.

The effect of modification of cysteine by the adduct formation with ^●^NO in the ATP binding site was studied by converting Cys431 of p-gp (pdb ID 6C0V) that is present in the ATP binding site to NO-bound Cys. Protons were added, followed by the addition of counter ions, and the resulting structure was then solvated in a box of water with 50,515 molecules. Prior to equilibration, the system consisted with 169,842 atoms was subjected to the following steps: (a) over a nanosecond of constrained molecular dynamics (MD) with the protein heavy atoms constrained to the original positions with a force constant of 10 kcal/mol/nm, (b) minimization, (c) low-temperature, constant-pressure MD simulation to assure a reasonable starting density, (d) minimization, (e) stepwise heating MD at constant volume, and (f) a constant-volume MD run for another 15 nanoseconds with incrementally decreasing the force constants. An equilibrated solution structure was then evaluated from a subsequent unconstrained 400-ns MD trajectory calculated at 300 K under constant volume (with 1 fs time step) using the PMEMD module of the Amber 18 program [30] to accommodate long-range interactions. All standard amino acid parameters were taken from the Amber SB14 force field. Atomic charges on NO-modified Cys were derived using the ChelpG scheme with the 6–31g(d) basis, set at the HF level using the program Gaussian 09.D01. An optimized structure of BCRP dimer was constructed (using pdb ID: 6HBU) by modeling the missing loop segments and carrying out the minimization under the Amber SB14 force field with two cysteine residues in the vicinity of the ATP binding site muted to NO-Cys.

### 2.7. Statistical Analysis

All experiments were carried out at least 3 times (*n* = 3) in duplicate. One-way analysis of variance (ANOVA) was used for statistical analysis using Graph pad prism (GraphPad Software, Inc, La Jolla, CA, USA). The Tukey Multiple comparison’s test was used for multiple comparisons. The results are expressed as mean ± SEM. The differences were considered statistically significant when *p* values were less than 0.05.

## 3. Results

### 3.1. Cytotoxicity Studies

Molsidomine: Molsidomine (MS), has been used extensively as a vasodilator that acts as a nitric oxide donor and has been used in angina pectoris [31], heart failure [32] and after myocardial infarction [33]. MS requires metabolic activation in liver to linsidomine (SIN-1) to generate NO. Cytotoxicity studies (Figure 2) indicated that MS was highly resistant to both NCI-RES/ADR cells (a P-gp expressing cell line) MXR cells (BCRP-expressing cell line).

Cytotoxicity of SIN-1, the active metabolite of MS, was also examined in these cell lines. In contrast to MS, our studies indicated that NCI/ADR-RES cell line showed no significant resistance to SIN-1, while the BCRP-expressing cell line was marginally (5–10-fold) resistance to SIN-1 (Figure 2).

### 3.2. Cytotoxicity of s-Nitrosoglutathione

S-nitroso glutathione is considered to be a source or reserve nitric oxide donor in vivo [34]. Studies indicated that both P-gp- and BCRP-expressing cells were resistant to GSNO (Figure 3). It is interesting to note that GSNO displaced biphasic effects in NCI/ADR-RES cell lines: cells were extremely resistant to low concentrations of GSNO while these cells were as sensitive as the WT cells at higher concentrations (10^−3^ M) of GSNO (Figure 3A). MXR cells were extremely resistant to GSNO and no biphasic effects were observed in BCRP-expressing cells (Figure 3B).

In order to confirm that cellular GSNO resistance is due to fact that GSNO may be a substrate for both P-gp and BCRP, we used verapamil and Ko143, known inhibitors for P-gp [35,36] and BCRP [37,38], respectively to examine GSNO cytotoxicity in these tumor cell lines. Data shown in Figure 4A clearly shows that GSNO is substrate for BCRP as Ko143 completely reversed GSNO resistance in MXR cell line. Ko143 had no significant effects on the cytotoxicity of GSNO in WT cells (Figure 4B). In contrast, verapamil (10 µM) only had a small (but significant) effect on GSNO cytotoxicity in NCI/ADR-RES cells (Figure 4C).

### 3.3. Cytotoxicity Studies with NCX4040

Several studies have indicated that NCX4040 is significantly cytotoxic to many different tumors in vitro [22]. Furthermore, NCX4040 have been reported to sensitize drug-resistance tumor cells to cis-platin in vitro and in xenografts [23]. In light of these studies, NCX4040 was utilized for reversal of drug resistance in MDR cells. While both NCI-RES/ADR cells and MXR cells showed some resistance to the lower drug concentrations NCX4040, both cell lines showed no significant resistance at higher concentrations of NCX4040 and IC_50_ values were very similar to their corresponding WT cells (Figure 5A,B). Both verapamil (10 µM) and Ko143 (0.5 µM) significantly modulated cytotoxicity of NCX4040 in NCI/ADR-RES and MXR cells, respectively, suggesting that at low concentrations, NCX4040 is a substrate for both P-gp and BCRP (Figure 5C,D).

### 3.4. Reversal of Drug Resistance by NCX4040

Our studies indicated that MS, SIN1 and GSNO were not significantly cytotoxic to these MDR cells and the fact that they were also substrates for both P-gp and BCRP, we utilized only NCX4040 for the further studies. While NCX4040 also appeared to be substrate for both P-gp and BCRP proteins at lower concentrations, NCX4040 still showed significant cytotoxicity in these MDR cells. Therefore, we examined if NCX400 could sensitize these MDR cells against ADR and TPT, in NCI/ADR-RES and MXR cells, respectively. Figure 6 shows that a subtoxic concentration of NCX4040 (1–2.0 µM, 10–20% cell death) was effective in modulating ADR and TPT cytotoxicity in NCI/ADR-RES and MXR cells. In contrast, NCX4040 had no effect on cytotoxicity of either OVCAR WT or MCF-7 WT cells (Figure 6).

As we found that at low concentration, NCX4040 was substrate of both P-gp and BCRP, and that it was still able to sensitize these cells to ADR and TPT, it is possible that at low ^●^NO concentrations ATPase functions of P-gp and BCRP was only partially inhibited. In order to confirm this possibility, we used higher concentrations of NCX4040 and examined reversal of ADR and TPT resistance. As shown in Figure 7, higher concentration (5 µM) was significantly more effective in reversing drug resistance, indicating that higher concentrations of ^●^NO was inhibiting ATPase activities of P-gp and BCRP.

### 3.5. Effects of NCX4040 on Cellular Drug Accumulation

ABC transporters significantly decrease intracellular concentrations of dugs as their main mechanism of resistance, effects of NCX4040 were examined on cellular accumulations of both ADR and Hoechst 33422 in respective NCI/ADR-RES and MCF-7/MXR cells. Hoechst 33422 accumulation has been used effectively to study functions of BCRP [29,39]. A small but significant increase in Hoechst 33422 accumulation was observed by NCX4040. Similarly, NCX4040 also enhanced accumulation of ADR in NCI/ADR-RES cells. However, these effects were significantly smaller compared to either verapamil or Ko143. NCX4040 had no effects on cellular accumulations of either ADR or Hoechst 33422 in WT cells (Figure 8).

### 3.6. Effects OF NCX4040 ON P-gp and BCRP Proteins

Decreases in either activities or expression of proteins are known to effect degree/fold drug resistance in tumor cells mediated by ABC transporters. It is, therefore, possible that NCX4040 or ^●^NO generated from it significantly modulated protein expression for P-gp and BCRP for their reversal of drug resistance in ABC-expressing tumor cells. We evaluated these possibilities by examining the effects of NCX4040 on the total protein levels with time. Our results presented in Figure 9 clearly show that ^●^NO/NCX4040 did not significantly modulate the proteins expression for either P-gp or BCRP in NCI/ADR-RES or MCF-7/MXR cell lines, indicating that modulation of protein expressions of P-gp and BCRP was not involved in the reversal of drug resistance by NCX4040 in these MDR cells.

### 3.7. Molecular Docking Studies

Scoring and Binding of NO-donors at the ATP Site: Binding of NCX4040 and various other NO-donors were calculated and compared to ATP binding to both unmodified and the modified to S-NO-Cys of the ATPase in both P-gp and BCRP. Furthermore, binding and stabilization of NCX4040 were also examined by docking NCX4040 to various conformations of the transmembrane domains of transporters as described previously [40,41]. As shown in Table 1, NCX4040 binding to these sites is similar in both P-gp and BCRP; however, it is relatively weaker than ATP for the unmodified ATPase in the ATP binding site. However, the binding score is similar to ATP when it binds to the modified S-NO-Cys-ATPase. It is interesting to note that S-nitroso glutathione showed very comparable binding scores as ATP for the S-NO-Cys site of the ATPase; however, it is significantly less cytotoxic to ABC transporter-expressing cells than NCX4040. As shown in Figure 10A,B, binding of NCX4040 is significantly stabilized in P-gp and BCRP proteins by both H-bonding and interactions with various amino acids in the nucleotide binding sites. Furthermore, the decreased binding of ATP as shown in Table 1 appears to be due to a significant steric hindrance from the bound NCX4040 (Figure 10) in both P-gp and BCRP at the unmodified and modified sites of the ATPase domain.

## 4. Discussion

Reduced intracellular drug concentrations mediated by ATP-dependent efflux proteins have been implicated in the development of multidrug resistance tumor cells both in laboratory and in the clinic. Overexpression of these efflux proteins, commonly known as ABC transporters, results in the cellular removal of chemotherapeutics, e.g., adriamycin, topotecan and mitoxantrone. This process requires binding of ATP to ATPase of the efflux proteins. Overexpression of these transporters has been detected in various cancer cells from patients undergoing treatment with chemotherapeutic drugs and presence of these proteins have been shown to cause treatment failures, leading to decreased survival. Furthermore, treatment with chemotherapy drugs has been shown to induces emergence of cancer stem cells (CTC’s). Cancer stem cells are subgroup of tumor cells that escape/survive chemotherapy and have been reported to promote relapse as CTC’s are highly invasive and resistant to anticancer drugs as ABC transporters are also highly overexpressed in CTC’s.

Thus, it is clear that in order to achieve cure for cancer one has to design and develop better anticancer drugs or alternatively find better approaches to deliver cancer drug to cellular targets in hope of overcoming emergence of drug resistant tumor cells. As the efflux proteins work to remove various anticancer agents from their targets in tumor cells using ATP/ATPase system, inhibition of ATPase functions in MDR tumor cells appears to be a natural cellular target for overcoming drug resistance in the clinic. Additionally, during chemotherapy treatment-related emergence of CTC’s occurs and that CTC’s express ABC transporters to remove drugs using ATP/ATPase system. There are a number of small molecules (e.g., verapamil or Ko143) that have been found to overcome/reverse multidrug resistance in vitro [6,42] or inhibit ATPase functions of P-gp or BCRP [43,44,45]. Furthermore, some of these compounds have shown significant activity against both P-gp and BCRP, resulting in reversal/sensitization of anticancer drugs in vitro [45]. Unfortunately, there are no known compounds that are currently utilized in the clinic to overcome ABC transporter-meditated resistance.

Thus, it is clear that that better site-specific targeted drugs must be developed that can inhibit functions of ABC transporters at subtoxic doses. Since ATPase function(s) are necessary to remove drugs from cellular targets, design of inhibitors of ATPase activities is most promising. Our previous studies using nitric oxide donors as the source of ^●^NO have indicated that ^●^NO can effectively inhibit ATPase activities of P-gp and BCRP in multidrug resistant tumor cell lines, resulting in sensitization of clinically active anticancer agents [10,11]. We proposed that this inhibition of ATPase functions results from nitrosation of -SH group of ATPase in the nucleotide binding site (s) of transporter proteins, resulting in decreased binding of ATP. ^●^NO is known to nitrosate various proteins [46], and we have shown that it nitrosates topoisomerase II [18,47], inhibiting its functions.

In our earlier studies we used DETANO (diethylenetriamine nonoate) and JS-K, a specific NO-donor that gets activated by GSH/GST systems in cells to generate ^●^NO [19], for the reversal of resistance in P-gp- and BCRP-expressing cell lines. While JSK was effective only in P-gp-expressing cells, it was found to be ineffective in BCRP-expressing cells. DETANO was effective in inhibiting ATPase activities of both P-gp- and BCRP-expressing cells, however, much higher concentrations of the donor were required for this reversal of drug resistance. Since NO-donors are also toxic to humans and DETANO is not tumor specific, we have now carried out further studies with other NO-donors in order to find a more universally effective inhibitor of ATPase activities in MDR tumor cells. Here, we found that NCX4040, a NO-donor related to aspirin, inhibited ATPase activities in both P-gp and BCRP-expressing cells and was also effective in reversing resistance of ADR and topotecan at subtoxic doses in MDR cells. In contrast, both S-nitroso glutathione and molsidomine were found to be substrates of P-gp and BCRP (Figure 2, Figure 3 and Figure 4), SIN1 was marginally active in both P-gp- and BCRP-expressing cells. Due to their extreme resistance and low cytotoxicity, these NO-donors were not used in any drug resistance reversal studies.

We found that NCX4040 was significantly cytotoxic to both sensitive MCF-7 and MXR cells and OVCAR-8 and NCI/ADR-RES cells with similar IC_50_ values. It is interesting to note that NCX4040 exhibited dual effects against both P-gp- and BCRP-expressing cell lines: at very low concentrations it showed resistance while at higher concentration it was as effective as sensitive cell lines. We also found that NCX4040 resistance at low concentrations was reversed by verapamil (P-gp) and Ko143 (BCRP) confirming that NCX4040 at low concentrations was substrates for both P-gp and BCRP. P-gp has been implicated in NCX4040 resistance in bladder carcinoma cell lines [20]. At higher concentrations of NCX4040, more ^●^NO is formed and released in these MDR cells leading to inhibition of ATPase activities of the transporters, resulting in enhanced cytotoxicity. This is further supported by our observations that 5 µM NCX4040 was significantly more effective in reversing resistance of both ADR and topotecan in P-gp- and BCRP-expressing cells than 2.0 µM NCX4040 (Figure 8).

While there were small changes noted, NCX4040 had no significant effects on the expression of either P-gp or BCRP proteins, indicating that increased cytotoxicity of ADR and topotecan was not due to modulations in the expression of ABC transporters but it resulted from significant inhibition of the transporter functions by NCX4040. NCX4040 increased intracellular drug concentrations in MDR cells, it was significantly smaller compared to either verapamil or Ko143 induced cellular accumulations in MDR cells. However, it is possible that cellular drug accumulation is time dependent and increases with time which was not studied here.

Our previous studies have suggested that ^●^NO generated via NO-donors are capable of inhibiting the ATPase activities of both P-gp and BCRP [10,11]. While we found that the inhibition of the ATPase activities of BCRP required significantly higher ^●^NO concentrations which may have resulted from the fact that BCRP does not contain -SH groups in its ATP binding site, we suggested that there are -SH functionality in periphery which can be successfully nitrosalated, resulting in a steric hinderance and decreased ATP binding. The ATP binding site of the ABC transporters appears to be critical for designing successful inhibitors of the transporters for the treatment of MDR in the clinic. While NCX4040 may work by nitrosation of ATPase -SH functions for observed reversal of MDR in both NCI/ADR-RES and MXR cell lines, other mechanisms cannot be ruled out at this time. These include nitrosation of other cysteines located in transmembrane domain of P-gp or BCRP, e.g., Cyst 592 or Cyst 608 which are essential for dimerization of BCRP for its functions [8,48]. Furthermore, it is possible that NCX4040 binds at transmembrane domains and sterically decreases the binding of ATP. Formation of reactive metabolites of NCX4040 in tumor cells, e.g., a methide may also be involved in the reversal of drug resistance in these MDR cells [49,50]. Methide then can react with -SH molecules of P-gp or BCRP, resulting in a decreased/inhibition of ATP binding and hydrolysis necessary for the functions of the transporters. Further work is in progress to address these possibilities.

The mechanism(s) of NCX4040 cytotoxicity is not clear at this time. It has been suggested to involve in addition to NO-induced apoptosis to induction of cyclooxygenase pathways (COX1 and COX2) as well as depletion of glutathione [21,22,23]. Currently, attempts are underway to evaluate the mechanism of actions of NCX4040 in MDR cells, and our preliminary work indicates that NCX4040 induces significant apoptosis in MDR cells with concomitant release of GSH in dying cells. While ADR and topotecan are both known to generate free radicals in tumor cells [51,52,53,54] it is not known how they interact with NCX4040 for synergistic tumor cell killing. Free radicals deplete glutathione and hence may also participate in tumor cell killing by increasing oxidative stress. However, these need to be further evaluated. Based on our molecular docking studies, a number of NCX4040 derivatives have been designed for synthesis and future evaluation as superior inhibitors of ATPase functions of ABC transporters.

## 5. Conclusions

Our studies presented here show that NCX4040, a nitric oxide donor derived from aspirin, was significantly cytotoxic to both ABCB1 and ABCG2-expressing human MDR tumor cells. We found that NCX4040 and/or ^●^NO generated from it inhibited ABC transporter activities by inhibiting their ATPase functions, causing reversal of the adriamycin and topotecan resistance in human MDR tumor cells. We also found that ^●^NO/NCX4040 significantly enhanced drug accumulations in MDR tumor cells. Our studies indicate that appropriately designed tumor specific nitric oxide donors may be useful in the clinic for treating human tumors that overexpress ABC transporters (including cancer stem cells) or in other diseases where drug resistance may result from the presence of ATP-dependent efflux proteins.

## Figures and Tables

**Figure 1 cancers-13-01680-f001:**
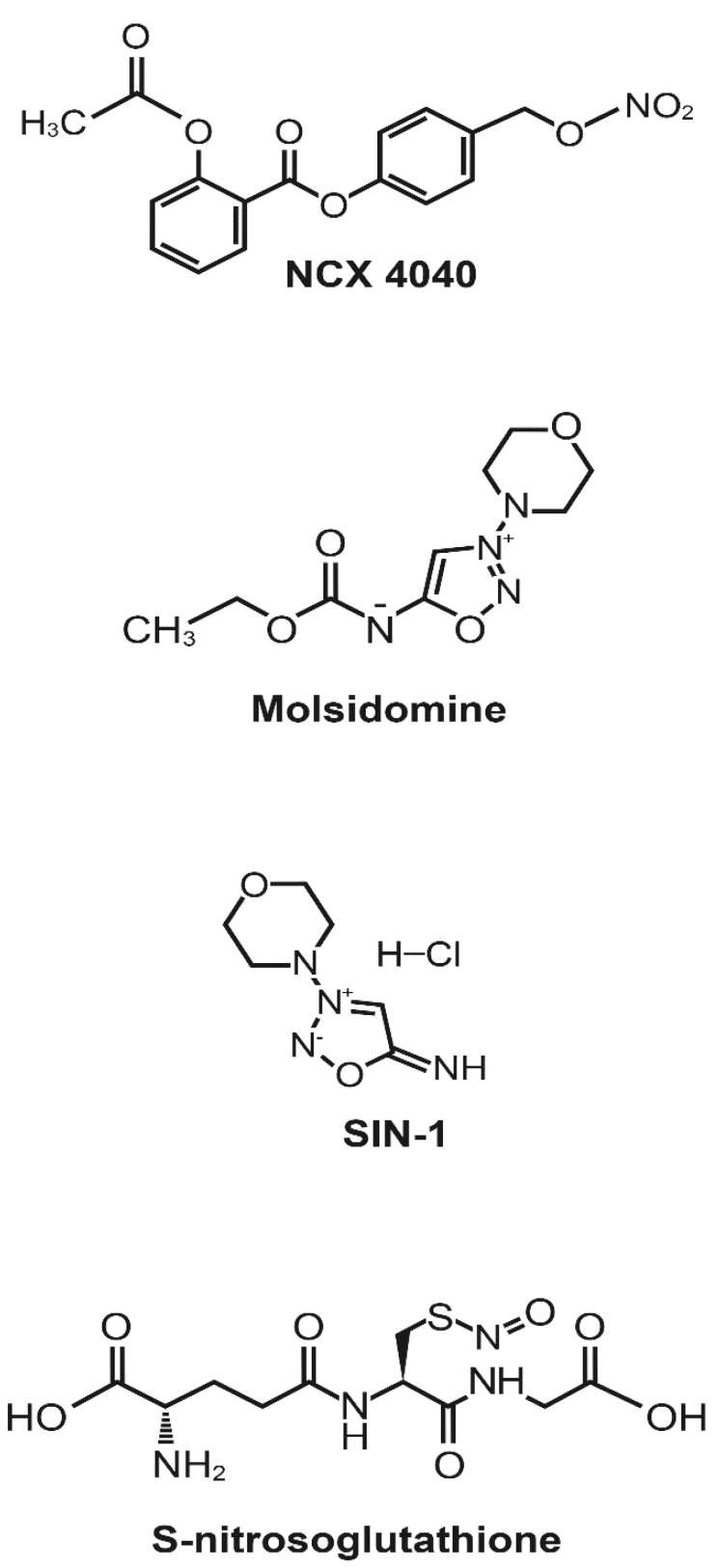
Structures of NCX4040 and NO-donors.

**Figure 2 cancers-13-01680-f002:**
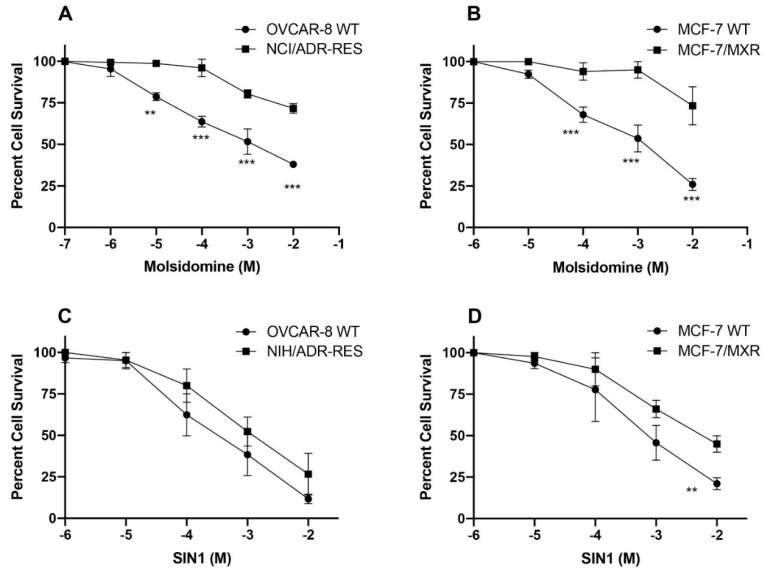
Cytotoxicity of MS in (**A**) WT OVCAR-8 and NIH/ADR-RES Cells; and (**B**) WT MCF-7 and MXR cells. Cytotoxicity of SIN-1 in (**C**) WT OVCAR-8 and NIH-RES/ADR Cells; and (**D**) WT MCF-7 and MXR cells. **, *** *p* values < 0.005, < 0.001 compared to concentration-matched samples.

**Figure 3 cancers-13-01680-f003:**
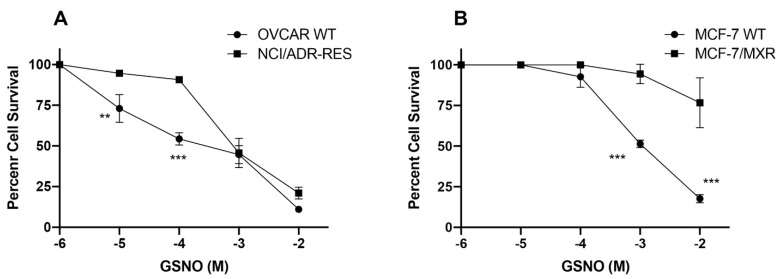
Cytotoxicity of GSNO in (**A**) WT OVCAR-8 and NIH-RES/ADR Cells; and (**B**) WT MCF-7 and MXR cells. **, *** *p* values < 0.005, <0.001 compared to concentration-matched samples.

**Figure 4 cancers-13-01680-f004:**
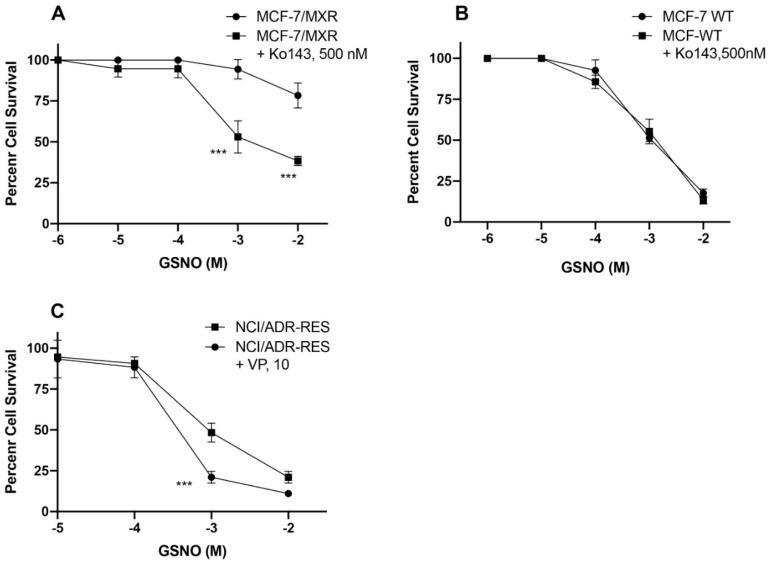
Cytotoxicity of GSNO in (**A**) MXR cells and (**B**) WT MCF-7 cells in the presence of 500 nm Ko143. Cytotoxicity of GSNO in (**C**) NCI/ADR-RES cells in the presence of Verapamil (10 µM). Ko143 or Verapamil was preincubated with cells for 2h in the complete medium before adding various concentrations of GSNO as detailed in the methods section. *** *p* values < 0.001 compared with concentration-matched samples.

**Figure 5 cancers-13-01680-f005:**
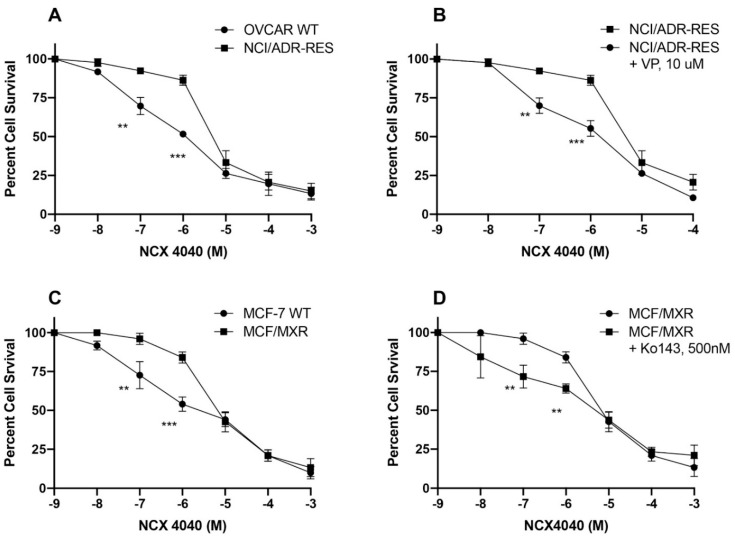
Cytotoxicity of NCX4040 in (**A**) WT OVCAR-8 and NCI-RES/ADR cells, (**C**) WT MCF-7 and MXR cells. Effects of verapamil (10 µM, (**B**)) and Ko143 (0.5 µM, (**D**)) on cytotoxicity of NCX4040 in NCI/ADR-RES and in MXR cells, respectively. ** and *** *p* values, <0.005 and <0.001, respectively compared to concentration-matched samples.

**Figure 6 cancers-13-01680-f006:**
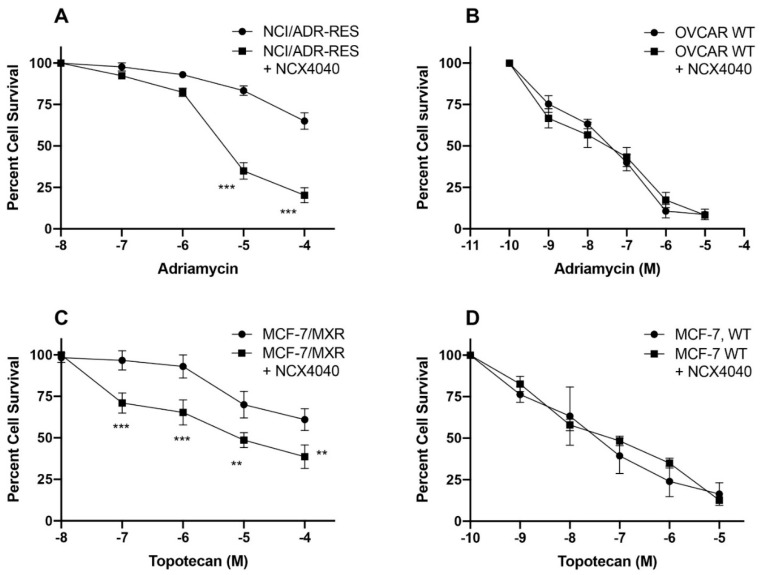
Reversal of adriamycin (**A**) and topotecan (**C**) resistance by NCX4040 (2 µM) in NCI/ADR-RES and MCF-7/MXR cells, respectively and effects of NCX 4040 (5 × 10^−8^M) in corresponding sensitive WTOVCAR-8 (**B**) and WT MCF-7 (**D**) cells. ** and *** *p* values < 0.005 and <0.001, respectively compared to concentration-matched samples.

**Figure 7 cancers-13-01680-f007:**
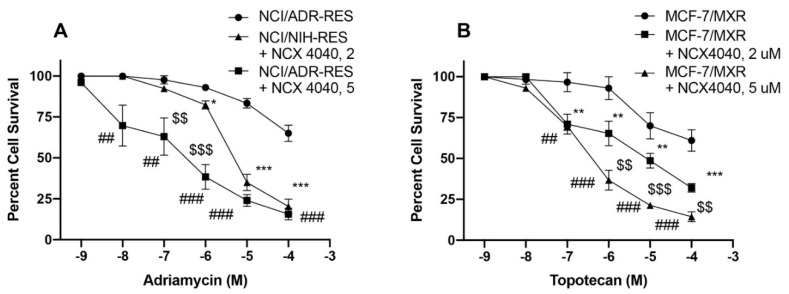
Reversal of adriamycin (**A**) and topotecan (**B**) resistance by NCX4040 (2 µM, and 5 µM) in NCI/ADR-RES and MCF-7/MXR cells, respectively. *, ** and *** *p* values 0.05, 0.005 and 0.001, respectively compared to concentration-matched samples. ##, ### *p* values <0.005 and <0.001 compared to drug alone and $$ and $$$ *p* values <0.005 and <0.001 compared to concentration-matched drug in the presence 2.0 µM NCX4040.

**Figure 8 cancers-13-01680-f008:**
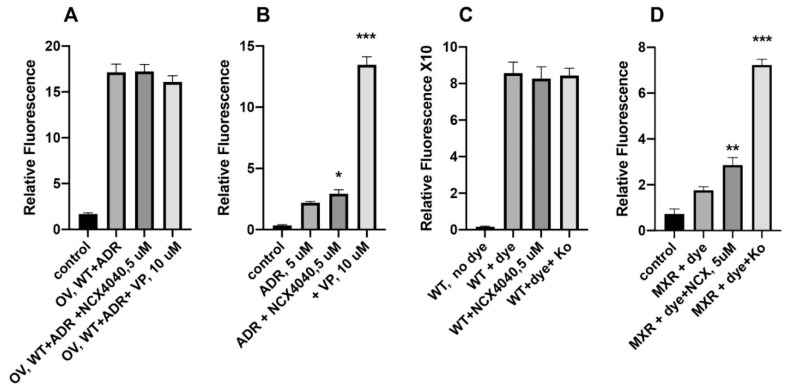
Effect of NCX4040 (5 µM) on accumulations of ADR in OVCAR WT (**A**) and NCI/ADR-RES cells (**B**) in the presence or absence of 10 µM verapamil. Effects of NCX4040 (5 µM) on accumulations of Hoechst 33342 in MCF-7 WT(**C**) and MCF-7/MXR cells (**D**) in the presence or absence of Ko143 (500 mM). The relative cellular fluorescence values in the presence of Ko143 × 10. *, ** and *** *p* values <0.05, <0.005 and <0.001, respectively compared to drug (dye) alone.

**Figure 9 cancers-13-01680-f009:**
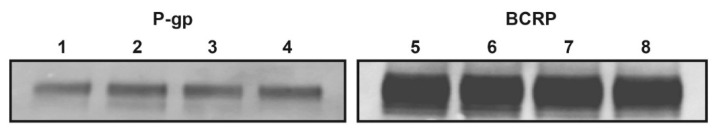
Effects of NO/NCX4040 (5 µM) on P-gp and BCRP proteins in NCI/ADR-RES and MCF-7/MXR cells following 4, 24 and 48 h treatment. Lanes 1 and 5 are untreated controls, 2 and 6 at 4 h; 3 and 7 at 24 h; and 4 and 8 at 48 h. A representative Western Blot is shown. The uncropped Western blots have been shown in Appendix A.

**Figure 10 cancers-13-01680-f010:**
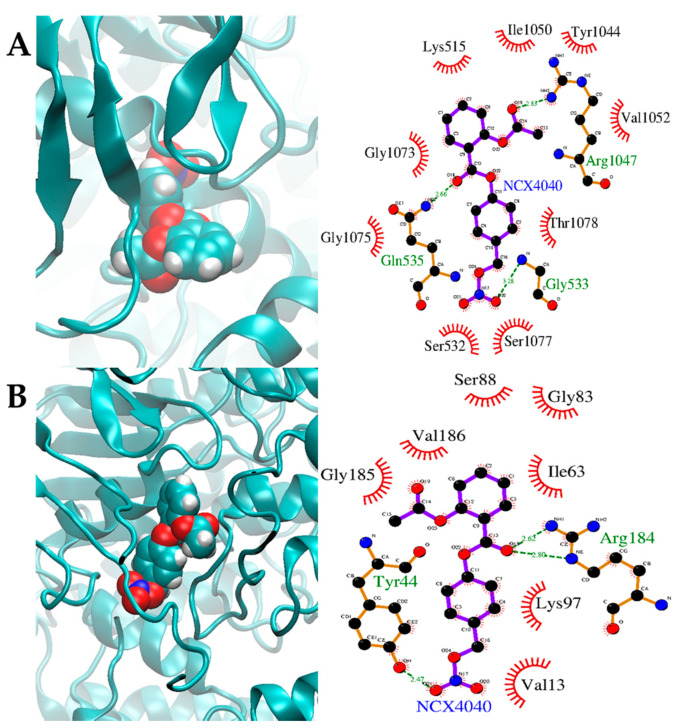
NCX4040 docked on (**A**) P-gp (pdb ID: 6C0V.pdb) (**B**) BCRP (pdb ID: 6hbu.pdb). NCX4040 is shown in solid spheres and the proteins are represented by ribbon diagrams. Residues of the protein that are in contact or making H-bonds are shown in right panels.

**Table 1 cancers-13-01680-t001:** Ligand interaction scores of NCX-4040, SIN1, S-Nitroso glutathione, and ATP. Contributions from shape, H-bond, protein desolvation, and ligand desolvation sum up to the total binding score. * Binding of ATP to the modified S-NO-Cys of the ATPase in P-gp, resulting from the reaction of NO generated from NO-donors as proposed previously [11].

Receptor	Ligand	Total	Shape	H-Bond	Protein Desolvation	Ligand Desolvation
P-gp	ATP	−15.2	−13.4	−20.4	6.9	11.7
	NCX-4040	−8.3	−11.7	−2.1	3.5	2.0
	S-Nitroso Glutathione	−8.9	−11.2	−8.1	3.9	6.5
	SIN1	−5.6	−6.0	−4.4	2.1	2.7
	ATP *	−7.2	−2.1	−10.1	7.5	12.0
BCRP	ATP	−13.2	−15.1	−19.5	7.7	13.5
	NCX-4040	−8.1	−11.8	−1.4	3.3	1.8
	S-Nitroso Glutathione	−9.1	−10.4	−8.0	3.0	6.3
	SIN1	−6.9	−7.2	−3.0	1.6	1.8

## Data Availability

No new data were created or analyzed in this study. Data sharing is not applicable to this article.

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
