# Peer review of "NCX-4040, a Unique Nitric Oxide Donor, Induces Reversal of Drug-Resistance in Both ABCB1- and ABCG2-Expressing Multidrug Human Cancer Cells"

_cancers, 2021, doi:10.3390/cancers13071680_

Round 1
Reviewer 1 Report
Sinha et al present a study of the role of several NO-producing compounds on the activity of ABC multidrug transporters. Studies on interventions that can inactivate or circumvent these transporters are of great interest in cancer therapy as these systems limit cancer cell sensitivity to a wide range of chemotherapy agents.
The study is based on previous work from the authors indicating that NO donors can inactivate these channels [Sinha et al. (2018) Nitric oxide reverses drug resistance by inhibiting ATPase activity of p-glycoprotein in human multi-drug resistant cancer cells, Biochim. Biophys. Acta 1862, 2806-2814; Sinha et al (2019) Reversal of drug resistance by JS-K and nitric oxide in ABCB1- and ABCG2-expressing multi-drug resistant human tumor cells, Biomed Pharmacother 120, 109468)]. The study itself is in general reasonably designed but there are several points that limit the study conclusions and should be addressed before publication
Major comments-
1) The authors combine in this work NO donors with very different profiles and jump between possible mechanisms of action such as nitrosation and direct binding to the receptor without further discussion. This is also exemplified in the use of NO-generating molecules with very different mechanisms of action that are not categorized in any way. Certainly, the underlying mechanism of inhibition can be complex to elucidate but in order to advance the field several possibilities should be discussed with different predicted effects, so experiments can be planned to prove or disprove the hypotheses. However, the authors present a very inconsistent discussion.
At the very least the following options should be discussed and evaluated in the light of the presented data – i) NO donors inactivate ABC channels by promoting S-nitrosation; ii) NO donors block the channels by directly binding to the channels; iii) NO inactivates the channels by secondary mechanisms that do not involve direct interactions of NO or the NO-generating molecules with the channels
2) The authors refer to the four molecules used as NO-donors; this suggests that all the molecules have similar mechanisms of action; this is not the case. NCX4040 is an NO-donor, when NO is released I can also serve as an inhibitor of cyclooxygenases. Molsidomine and SIN-1 are NO and superoxide donors and can generate peroxynitrite or NO depending on the presence of suitable electron acceptors (Singh et al (1999) The Peroxynitrite Generator, SIN-1, Becomes a Nitric Oxide Donor in the Presence of Electron Acceptors. Arch Biochem Biophys 361, 331-339). GSNO is mainly a transnitrosating agent that can react with thiolates to form glutathione and a nitrosothiol without the formation of any free NO (Hogg (1999) The kinetics of S-transnitrosation—a reversible second-order reaction. Analytical biochemistry, 272, 257-262). These differences are critical when trying to understand the mechanisms of action involved in the ABC transporter inactivation.
3) The molecular modeling as carried out is irrelevant. The authors indicate that the procedure is similar to other references [e.g. ref40 Alam et al. (2018) Structure of a zosuquidar and UIC2-bound human-mouse chimeric ABCB1, PNAS 115, E1973-E1982] but this is not the case. It is well established that ABC transporters bind the substrates to be transported in a cavity within the transmembrane domain [Dastvan et al., (2019) Mechanism of allosteric modulation of P-glycoprotein by transport substrates and inhibitors. Science 364, 689–692; Alam et al. (2019) Structural insight into substrate and inhibitor discrimination by human P-glycoprotein Science 363, 753–756; (2019) Alam et al. (2018) Structure of a zosuquidar and UIC2-bound human-mouse chimeric ABCB1, PNAS 115, E1973-E1982; Waghray & Zhang (2018) Inhibit or Evade Multidrug Resistance P‑Glycoprotein in Cancer Treatment J. Med. Chem. 61, 5108−5121]. Thus, in absence of compelling evidence suggesting otherwise, modeling studies should focus on this binding site. However, the authors are modeling the ligand in the ATPase domain, with no biochemical evidence of NCX4040 binding on this site. Even assuming that the molecules are interacting with the receptors in this location -which is very far from proven here- these studies indicate a lack of understanding of the possible mechanism. If NO or SNO is involved in the inactivation, why the NO-releasing molecules have to bind to the transporter in the ATP site? There is no need for that. Conversely, if the molecules inactivate the transporter by direct binding, NO is not involved in the process, contrary to the authors’ hypothesis
4) lines 386-387 the authors propose that NO nitrosates a thiol in the ATP binding site of the transporter. This is not supported by the data as ABCG2 could not be inactivated in this way. There is a Cys residue close to ATP in ABCB1 but not in ABCG2. Moreover, if this residue were easily inactivated by nitrosation then GSNO should be the most efficient inactivator from among the agents used here (if site access is a concern, perhaps Cys-NO could be studied as an alternate transnitrosating agent)
5) This manuscript, along with previous work from the authors seems to support a role for NO in transporter inactivation independently of nitrosation (otherwise GSNO should be as effective as NO donors inactivating the channels, or probably even better). There are many possibilities here, but for example NO can directly inhibit mitochondrial respiration and induce increased ROS production [Hortelano, S. et al (1997). Nitric oxide induces apoptosis via triggering mitochondrial permeability transition. FEBS letters, 410, 373-377; Wei, T. et al (2000). Nitric oxide induces oxidative stress and apoptosis in neuronal cells. Biochimica et Biophysica Acta, 1498, 72-79; Snyder, C. M. et al (2009). Nitric oxide induces cell death by regulating anti-apoptotic BCL-2 family members. PloS one, 4(9), e7059]. These mechanisms could be tested in the future using NO donors along with mitochondrial ROS scavengers such as mitoQ
Other points:
6) The authors use indistinctly P-gp and ABCB1 through the text; the same applies to BCRP and ABCG2. It will be less confusing if they stick to one of the names for each protein throughout the text.
7) DETNO is not a generic abbreviation for diethylenetriamine nonoate. DETANO or DETAnonoate is the common abbreviation.
8) Mentions of “S-nitrosylation” should be corrected to the more correct term “S-nitrosation” (Heinrich, T. A. et al (2013). Biological nitric oxide signalling: chemistry and terminology. British journal of pharmacology, 169, 1417-1429.)
9) In line 400; NCX4040 is misspelled
Reviewer 2 Report
In this article Sinha and co-authors described the use of NCX-4040 as NO-donor able to reverse drug resistance in ABCB1 and ABCG2 human cancer cells. In particular, the authors found that NCX-4040 is significantly cytotoxic to both sensitive MCF-7 and MXR cells and OVCAR-8 and NCI/ADR-RES cells. The authors sustain that NO released by NCX-4040 or NCX-4040 inhibit ABC transporter activities, but they never evaluate the NO released by the compound or the stability of NCX-4040 in cell medium.
The authors used Molsidomine and SIN-1 as NO-donors reference compounds; these two NO-donors are completely different in terms of lipophilicity, and mechanism of NO-release. SIN-1 is a spontaneous NO-donor and it is very hydrophilic, Molsidomine is a prodrug of SIN-1 and require a metabolic activation to release NO.
NCX-4040 is a nitrate ester and release NO with a different mechanism, probably require the presence of aldehyde dehydrogenase to release NO. NCX-4040 is an acetyl salicylic acid derivative, it is an ester, so it is metabolically instable and it is known that in the cells the carboxylic ester moiety is hydrolysed (Journal of Medicinal Chemistry, 2007, Vol. 50, No. 10). After hydrolysis the phenolate intermediate undergo a 1,6-elimination with formation of a quinone methide, a very good electrophile that could react with numerous nucleophile present in the cells, like glutathione. So the cytotoxic effect of NCX-4040 could be due to NO release or to the formation of the reactive quinone methide.
Due to the instability in the cells of NCX-4040 the authors cannot be sure that is NCX-4040 that interact with P-gp and that NCX-4040 enhance accumulation of ADR.
The authors should demonstrate:
- that in their test NCX-4040 release NO, they should repeat the test in the presence of a NO scavenger;
- that the cytotoxicity of NCX-4040 is due to NO released and not to the formation of quinone methide;
- that the enhance of accumulation of ADR is due to NCX-4040 and not to the formation of quinone methide.
The article should not be published in the present form but after major revision.
NCX-4040 and NCX-4016 are widely tested as antitumor agents, especially against colon cancer cells, but the mechanism of their activity was never completely understood.
Round 2
Reviewer 2 Report
In this article Sinha and co-workers aim to identify a NO-donor that can inhibit ATPase activities of ABC transporter at a subtoxic dose.
NCX-4040 is a NO-donor able to release NO in tumor cells as demonstrated previously, but the authors have not demonstrated that the ability of NCX-4040 to inhibit ABC transpoters is NO-dependent. In the discussion of the results the authors, correctly, suggest that the inhibition of the ATPase activities could be due to NO, or to thiols depletion, or to NCX-4040 metabolites.
In my opinion the paper should be published in the present form, but the authors should modify the summary where they wrote "In this study we have found that ●NO 16 generated from NCX4040, a nitric oxide donor, inhibited ABC transporter activities by inhibiting 17 their ATPase functions, causing reversal of both adriamycin and topotecan resistance in human 18 MDR tumor cells. We also found that NO/NCX4040 significantly enhanced drug accumulations in 19 MDR tumor cells. These studies strongly suggest that tumor specific nitric oxide donors that deliver 20 high amounts of nitric oxide to clinical resistant tumors may be extremely useful in treating human 21 tumors overexpressing ABC transporters, including cancer stem cells." The authors should chenge this part and also the abstract, otherwise could seems that the ATP inhibition was due only to nitric oxide released.
Author Response
I have modified/changed the summary and abstract as indicated in red.